# Effect of the Intake of Lean Red-Meat from Beef-(Pirenaica Breed) versus Lean White-Meat on Body Composition, Fatty Acids Profile and Cardiovascular Risk Indicators: A Randomized Cross-Over Study in Healthy Young Adults

**DOI:** 10.3390/nu14183724

**Published:** 2022-09-09

**Authors:** Alba M. Santaliestra-Pasías, María L. Miguel-Berges, María M. Campo, Ana Guerrero, José Luis Olleta, Pilar Santolaria, Luis A. Moreno

**Affiliations:** 1GENUD (Growth, Exercise, Nutrition and Development) Research Group, Facultad de Ciencias de la Salud, Universidad de Zaragoza, 50009 Zaragoza, Spain; 2Instituto Agroalimentario de Aragón (IA2), Universidad de Zaragoza-CITA, 50001 Zaragoza, Spain; 3Instituto de Investigación Sanitaria Aragón (IIS Aragón), 50009 Zaragoza, Spain; 4Centro de Investigación Biomédica en Red de Fisiopatología de la Obesidad y Nutrición (CIBERObn), Instituto de Salud Carlos III, 28220 Madrid, Spain; 5Departamento de Producción Animal y Ciencia de los Alimentos, Facultad de Veterinaria, Universidad de Zaragoza, 50009 Zaragoza, Spain; 6Departamento de Producción Animal y Ciencia de los Alimentos, Escuela Politécnica Superior de Huesca, Instituto Universitario Ciencias Ambientales, Universidad de Zaragoza, 22071 Huesca, Spain

**Keywords:** meat intervention, cardiovascular profile, diet, body composition, young adult

## Abstract

The main dietary guidelines recommended a restriction of total and saturated fat intake in the management of cardiovascular risk. These recommendations are overgeneralized, and all red meats should be limited and replaced by white meat. The aim is to assess the effect of the consumption of beef (from the Pirenaica breed) or chicken-based diets on body composition, fatty acid profile and cardiovascular (CV) risk indicators in healthy adults. A randomized cross-over study was carried out in three University accommodation halls. Participants consumed either the Pirenaica breed beef or chicken three times per week for 8-week periods with their usual diet. Body composition, clinical, biochemical and dietary variables were evaluated at baseline and at the end of each period. A validated diet questionnaire was used to assess nutrient intake and monitor compliance. Intervention and control group comparisons were done with the general linear regression model for repeated measures. Forty-seven healthy adults were included (51.6% males, mean age 19.9 ± 1.75 years). No significant differences were found in body composition, fatty acid profile or CV risk indicators from baseline in either diet group. Consumption of lean red meat (Pirenaica breed) or lean white meat (chicken) as part of the usual diet is associated with a similar response. Clinical Trial Registration: NCT 04832217 (accessed on 6 September 2022).

## 1. Introduction

The main dietary guidelines for the prevention of cardiovascular diseases (CVD) recommend the restriction of total fat to less than 30% of energy intake, including a restriction of saturated fatty acids (SFA) to less than 10% [1]. However, the guidelines oversimplified the recommendations regarding low SFA intake. In this sense, red meat is one of the main restricted foods because of its richness in total fat and SFA [2]; however, the fat content between different types of red meat is extremely variable. The source of animal, breed, feeding, processing and the particular cut of the meat have an influence on the quantity and quality of the product, and also the cooking method will influence the final nutrition content [3,4,5], as was also observed in lamb [6,7].

Current recommendations to limit red meat and processed meat are based on the relationship with the high risk of most important non-communicable chronic diseases (i.e., diabetes, gastrointestinal cancer, CVD, etc.) [8,9,10]. Moreover, the International Agency for Research on Cancer (IARC), the cancer agency of the World Health Organization, has classified processed meat as carcinogenic to humans and red meat as probably carcinogenic to humans [11]. The results of some meta-analyses (including case control and cross-sectional studies) are more susceptible to recall and selection biases (i.e., dietary recall bias) than cohort studies [12]. Moreover, the high heterogeneity of observational studies, in terms of study design, sample size, demographic factors, intake unit or serving sizes, and confounders, among others, could limit the results’ interpretation on the associations between meat and diseases [13]. In fact, the current dietary guidelines for processed meat and non-processed red meat for adults suggest the limitation of non-processed red meat consumption (weak recommendations, low-certainty evidence) and processed meat consumption (weak recommendations, low-certainty evidence) without taking into consideration animal welfare and environmental issues [14]. On the other hand, individuals with a high intake of red meat and processed meat usually practice a general unhealthy lifestyle behavior with, for example, low consumption of fruits, vegetables and whole grains, smoking or being less physically active, among others [15]. All these factors should be taken into consideration regarding dietary guidelines as a whole.

The development of some specific regions, such as the Pyrenees Mountains, includes the conservation of the environment, taking into consideration local agriculture and farming practices. The special situation of these mountainous husbandry systems offers, to livestock rearing, a favorable atmosphere to develop the current kind of meat, especially for extensive beef livestock that could develop a commercial label (Ternera de los Pirineos, Pirenaica breed). This breed is a fast-growing beef breed [16] producing meat with a low percentage of intramuscular fat [17]. These animals are often reared and fattened in communal pastures at the end of spring and summer. In addition, consumers reveal that regional origin is one of the most relevant factors at the moment of choice for the type of veal, especially among young consumers [18]. Therefore, the aim of the current study is to evaluate the specific effect of the consumption of the Pirenaica breed in healthy adults on their body composition, cardiovascular risk indicators and fatty acid profiles compared with the consumption of conventional chicken.

## 2. Materials and Methods

### 2.1. Study Design

The current study is a cross-over randomized controlled study (NCT 04832217) developed in the framework of the DIETAPYR2 study (*Innovaciones aplicadas a la cadena productiva pirenaica de vacuno*
*para valorizar una carne identificable por el consumidor*). It was developed into three university accommodation halls, two of them in the city of Huesca (*Colegio Mayor Universitario Ramón Acín and Residencia Misioneras del Pilar*) and one in the city of Zaragoza (*Residencia Baltasar Gracián*) (Zaragoza, Spain). The study consisted of two experimental periods with a duration of 8 weeks, respectively, with a washout period between them of 5 weeks. Enrolled participants were randomly assigned to a beef (Pirenaica breed) or a conventional chicken-based group. The washout period took place after each period to remove the possible residual effects of the preceding experimental diet on the different variables tested. Participants were instructed to maintain a healthy diet and were asked not to change their diet or activity habits. The flow diagram with the recruitment procedure is presented in Figure 1.

The study design is presented in Figure 2. The first visit was scheduled for the previous afternoon, where medical history and questionnaires (sociodemographic and lifestyle behaviors) were applied, also anthropometric, blood pressure (systolic and diastolic blood pressure) and heart rate measurements were taken. Additionally, after an overnight 12-h fast, first blood drawn was taken to assess the cardiovascular profile, hematology, iron metabolism, other minerals (calcium, magnesium and zinc), apolipoproteins (A1 and B), reactive protein C and fatty acid profiles. Stool samples were also collected.

At the time of the first visit, each participant was randomly assigned to a beef (Pirenaica breed) or a chicken-based diet (intervention or control group, respectively).

After 8 weeks, participants attended a second visit in the afternoon and the second assessment of the blood extraction and stool sample the following day in the same conditions. Following the cross-over design, participants for the second 8-week period were crossed over to the beef (Pirenaica breed) or chicken-based diet, respectively. The clinical study with all the measurements was done again at the beginning and at the end of the second 8-week period, separating both periods with a 5-week wash out period.

The study design and duration of the study were based on previous research [19]. Moreover, the population group (young healthy adults) implies the need to work in university accommodation halls to facilitate the consumption of food in a standardized way (frequency of food, cooking methods, etc.). Moreover, university courses in Spain are based on three trimesters, and we need to adapt the study design with university periods to perform the clinical study in the periods of attendance of participants and also to guarantee the consumption of products. Additionally, residences give us the opportunity to control the consumption of products (chicken or the Pirenaica breed at lunch time) by the accommodation hall’s personnel which controlled the consumption of the corresponding chicken or Pirenaica breed. Also, the control of the number of times that each participant consumed their corresponding meat was registered.

### 2.2. Participants

Assuming a two-tailed alpha error of 0.05, with a statistical power of 90% and dropout rates of up to 20%, the required sample size was 60 participants in total, 30 per group. Study participants were randomized 1:1 into two equally sized groups. Computer-generated random allocation was centrally elaborated and stratified by sex. The procedure was internet-based and developed by one investigator of the team (MLMB). Another investigator was responsible for enrolling participants (AMSP).

Participants recruited were older than 18 years (age range: 18.1 to 27.5 years of age) and Caucasian. Despite all the efforts, 52 participants agreed to participate in the study, and before starting the baseline, two refused to participate, and three did not accept to participate in the second one. The recruitment periods involved the 3 previous months before the start of the baseline period. Finally, a total of 47 participants (51.1% males) were included. Eligibility criteria included being free of any chronic, metabolic, endocrine or nutrition-related disease. In the medical history, participants were asked to report medical treatment. The limitation of the total sample size should be assumed due to the design of the study and the periods when the project should be performed in the university accommodation halls.

All subjects gave their informed consent for inclusion before they participated in the study. The study was approved by the Research Ethics Committee of the Government of Aragon (Spain) (N° 17/2018, 11 October 2018). The study was performed following the ethical guidelines of the Edinburgh revision of the 1964 Declaration of Helsinki (2000).

### 2.3. Socieconomic Indicators

Maternal and paternal education levels were used as indicators of socioeconomic status (SES). For both questions, eight response categories were offered, and were categorized into three groups: (1) Low SES: Basic studies; (2) Medium SES: Professional formation; (3) High SES: University studies.

### 2.4. Assessment of Anthropometrics and Lifestyle Behaviors

Anthropometric measurements were always obtained by the same trained researcher. Body weight (kg) was measured with an electronic scale (Tanita BC 418 MA, Tanita Europe GmbH, Sindelfingen, Germany). The participant stood on the platform of the scale without support, with the body weight evenly distributed between both feet. Light indoor clothing was worn, excluding shoes, long trousers and sweaters. Height (centimeters) was obtained with a precision stadiometer (SECA 225), with a precision of 0.1 cm and a range of 70–200 cm. The participant stood straight in an upright position: feet together, knees straight, buttocks and back directly touching the back part of the stadiometer. The head was positioned in the Frankfurt plane. The arms were relaxed on the side of the body, with the inner part of the hand facing the thigh. The mobile, horizontal part of the stadiometer touched the head of the participant with light pressure on the hair. Fat mass was obtained from bioelectrical impedance analysis (Tanita BC 418 MA, Tanita Europe GmbH, Sindelfingen, Germany). Body mass index (BMI) (kg/m^2^) and fat mass index (kg fat mass/m^2^) were calculated.

In addition, waist circumference (WC) as an indicator of abdominal fat was measured using an inelastic tape (SECA 200), range 0–150 cm, at the midpoint between the lowest rib margin and the iliac crest, near the level of the umbilicus, at the end of gentle expiration, with the subject in a standing position and recorded at the nearest 0.1 cm.

Blood pressure (BP) measurements were performed during the examination day in a quiet room using an automated oscillometer device (OMROM HEM-7051-E). Systolic BP and diastolic BP were measured at the right and left arms while the participant was in a seated position with the back supported, uncrossed legs, feet on the floor and the upper arm at heart level. Three measurements were taken at 2-min intervals. The mean of the three values was used for the analyses.

To assess dietary compliance, participants were asked to complete four food frequency questionnaires, at the beginning and at the end of each 8-week period. The questionnaire was previously validated [20,21]. Moreover, to assess the physical activity levels, the Adapted International Physical Activity Questionnaire (IPAQ-A) was applied [22]. Regarding sedentary behaviors, the questionnaire was based on the previously validated in the HELENA study [23].

### 2.5. Intervention

The participants were asked to consume their usual diet, including the corresponding chicken or Pirenaica breed products, three times per week. The menu between the 3 accommodation halls were similar, and no macro and micronutrient intakes were analysed. The nutritional values of chicken-based and the Pirenaica breed-based diets were similar, including sources of dietary proteins and fats. Participants following the study were instructed to consume 150 g (g) of boneless chicken or beef (200 g with bones). The beef was obtained from entire young bulls from the Pirenaica breed that had been reared grazing in extensive conditions until weaned at around 6 months, and fed cereal-based concentrates and cereal straw *ad libitum* afterwards until slaughtered at around 14 mo. The loin, silverside and brisket of the young bulls were used. To ensure harmonization, product-rich diets were served during lunch time and each chef of the designated university accommodation halls was given instructions on the cooking methods. The cooking methods are presented in Table 1.

### 2.6. Laboratory Analysis

Blood samples (total: 4 blood samples) were drawn via venipuncture after a 12 h overnight fast. Samples were immediately shipped to the laboratory. Standardized laboratory procedures were used to analyze samples for glucose, urea, creatinine, high density lipoprotein cholesterol (HDLc), low density lipoprotein cholesterol (LDLc), triglycerides, glutamyl oxaloacetic transaminase (GOT), glutamyl pyruvic transaminase (GPT), hemoglobin, hematocrit calcium, magnesium, zinc, apolipoproteins A1 and B, iron metabolism (iron, ferritin, transferrin and transferrin saturation index) and C reactive protein. Spectrophotometry, colorimetric enzymatic, colorimetric, immunochromatography and turbidimetric analyses were done via VITROS 5600. SYSMEX was also used to perform the hemogram, and the Architest Plus to perform the turbidimetric analysis for apolipoprotein A1 and B.

The fatty acid profiles were analyzed at the University of Granada (Spain). Plasma fatty acids were quantified after methylation by gas-liquid chromatography coupled with mass spectrometry detection (GS-MS). Fatty acid methyl esters from plasma lipids were methylated and extracted as previously reported by Lepage & Roy [24].

The included fatty acids analyzed were the following: Palmitic Acid, Stearic Acid, Oleic Acid, Linoleic Acid, Arachidonic Acid (AA_W6), Eicosapentaenoic Acid (EPA), Docosahexaenoic Acid (DHA), Saturated fatty acids (SFA), Unsaturated fatty acids (UFA), monounsaturated fatty acids (MUFA), diunsaturated fatty acids (DUFA), ratio MUFA/DUFA, polyunsaturated fatty acids (PUFA), ratio MUFA/PUFA, PUFA *n*6, PUFA *n*3, PUFA > 18C n6 and PUFA > 18C *n*3, ratio SFA/MUFA.

### 2.7. Statistical Analysis

Descriptive study characteristics are shown as mean and standard deviation for continuous variables and number of cases and percentages for categorical variables. The validity of the cross-over designs (i.e., the absence of a carry-over, period effect and interaction) was tested by a repeated measures model (Analysis of Variance, ANOVA), defining a one-two-level model, where the order of treatment was the between-participants factor and the differences in the dependent variables were the within-participants’ indicators. When significant differences were detected, multiple comparisons with *t*-tests were made using the Bonferroni correction for normally distributed variables or the Wilcoson test for paired samples for variables with a skewed distribution to check between which periods differences occurred. In those parameters in which multiple comparisons using Bonferroni were performed, a significance level corrected by Bonferroni (*p* value/combination number; 0.05/46 = 0.001087) has been used because of this test’s influence in sample size calculations; therefore, statistical significance was considered when *p* < 0.001087. The significance level was set at *p* < 0.05 for the rest of the variables. In the parameters in which one or more of the three effects were not satisfied, only the results of the first period are presented. The results of blood pressure (systolic and diastolic pressure), hemoglobin, hematocrit, glucose, glutamic pyruvic transaminase (GPT), gamma-glutamyl transferase (GGT), total cholesterol, LDLc, ratio LDL/HDL, ferritin, transferrin, magnesium, calcium, apolipoprotein B, C reactive protein (CRP) and linolenic acid were only from the first intervention period.

Intervention and control groups comparisons, i.e., the beginning and the end of the chicken and beef (Pirenaica breed) groups, and also the mean difference between the beginning and the end of both groups were done with the general linear regression model for repeated measures.

All analyses were done using the Statistical Package for the Social Sciences (SPSS Version 21 for Windows; SPSS, Chicago, IL, USA).

## 3. Results

From the total 52 participants who agreed to participate, 47 (87.0%) finalized the complete study with an acceptable compliance of follow-up (did not follow the diet exactly as offered in the accommodation halls, but made an acceptable modification to the diet). In these circumstances, registered dietitians helped to increase compliance of the study’s requirements. Table 2 shows the sociodemographic characteristics of participants, and also the baseline blood sample parameters. In Table 3, body composition and blood parameter indicators, with the mean before and after each intervention period, and also the mean differences over time between intervention periods from the total sample, were included. In the chicken diet group only, statistical differences in the arachidonic acid and the total PUFA (>18C *n*6) levels were observed (*p* = 0.002), showing a decrease in both fatty acid levels. In the Pirenaica breed beef diet group, triglycerides and oleic acid showed a statistical increase during these periods; however, a decreased level in the transferrin saturation index, arachidonic acid and also the total PUFA (>18C *n*6) were observed. When analyzing the mean differences over time in both interventions, chicken versus Pirenaica breed diets, no statistical differences were observed in any of the body compositions or blood sample parameters over time.

In Table 4, the results of the variables that did not comply with the carry over effect, the period effect or the interaction effect were included. For these reasons, the current table includes only the information from the first period of the project, the first 8 weeks of treatment (*n* = 23, chicken-based diet group; *n* = 24, Pirenaica breed beef-based diet group). The mean differences in each diet group and the mean difference over time between groups were included. Regarding the blood pressure, systolic BP increased in the chicken diet group (*p* = 0.023), however, no effect was observed when we analyzed the mean differences over time. Regarding the hematology or the lipid profile, in the chicken diet group, an increase in the glucose levels and an increase in the hemoglobin and also in GGT, which was also observed in the Pirenaica breed beef diet group, were statistically significant (*p* < 0.05). However, after checking the mean differences over time, no effect was observed. Considering the specific effect of the diet on the iron metabolism, a decrease in ferritin levels was observed in the Pirenaica breed group, and the transferrin levels increased in both diet groups (*p* < 0.05). A decrease in magnesium levels was observed in the chicken diet group, and a decrease in calcium levels was observed in both groups. Moreover, a decrease in the apoliprotein B levels and an increase in the CRP levels were observed in the chicken group. Nevertheless, none of these differences were statistically different after analyzing the mean differences over time.

Additional analyses were done by sex regarding the iron metabolism. These results can be found in Table 5. Iron and transferrin saturation index were analyzed in all samples in both periods; however, ferritin and transferrin levels were analyzed only in the first period of the study due to both of them failing to meet the carry-over effect. Ferritin levels decreased in both the male and female’ groups after consuming the Pirenaica breed beef diet, with a higher decrease in the male group (*p* < 0.001, *p* = 0.039, respectively). Regarding the transferrin levels, increased levels were observed in both gender groups, showing a higher increase in the female group (*p* < 0.001).

## 4. Discussion

Although red meat consumption has a controversial effect on several disease-risk indicators, the current results show that an intensive intervention with the inclusion of a three-days per week of lean Pirenaica breed (intervention) and a lean chicken-based diet (control), did not modify body composition, serum lipid profile or fatty acids parameters in institutionalized young, healthy adult participants. As no effect was observed with both meat interventions, the authors hypothesized that other associated factors could be behind the related effects between meat consumption and higher disease risk factors that were found in other published studies.

### 4.1. Meat Consumption and Dietary Guidelines

The excess of meat consumption above the current dietary guidelines is associated with a higher diet-related health risk [11]. Although red meat consumption during the study exceeded the present Spanish dietary recommendations [19], no effects in terms of body composition or other disease risk factors were observed. It is important to remark that total meat consumption at the beginning and end of both the beef- and chicken-based periods did not differ (*unpublished results*). Another feature that should be considered was the comparison effect of consumption below or above the recommendations because most of the published manuscripts compared the consumption by categories (lowest vs. highest, obtained by the media, median, tertiles, quartiles, or so on), and on most occasions, the highest category doubles or triples or even more, the amount consumed in the lowest category. One possible future approach could be to focus on the recommendations for meat consumption, or just above them, in order to check the potential health- or disease-related associations.

Balancing protein intake (plant and animal) across all food groups following the current dietary recommendations [19] could be the key in reducing associated risk, and not only focusing on meat reduction. In this sense, the animal sources of protein, such as meat, fish and eggs, and plant-based sources of proteins, such as legumes, should be included at the rate of 3–4 times/week for each group. The overall dietary pattern should include an equilibrium in sources of protein and selecting the best type of meat seems to be one acceptable recommendation, including high quality lean meat such as the lean Pirenaica breed.

### 4.2. Meat Consumption and Body Composition and CVD Risk Factors

Our results are in line with some previous results of intervention studies, focusing on the comparison between lean lamb and chicken consumption in institutionalized young populations [25] and elderly women [26]. No effects on the main body composition or cardiovascular disease indicators were observed. However, our results are in contrast with a recent meta-analysis of observational studies [27], which concluded that red meat consumers placed at the highest percentiles had a higher BMI and WC compared with those allocated to the lowest percentiles. Another meta-analysis of cohort studies [28] concluded that the consumption of red meat was positively associated with BMI and being overweight and obesity. However, no comparison was made between different types of meat. The associations with BMI were more evident in the US than in European cohorts (1.8 vs. 0.5 increase in BMI each 100 g/day of red meat), while no associations were found for Asian cohorts. No meta-analysis of intervention studies was found.

Another kind of study has focused on the diet’s fatty acid content, including several types of meat. A comparison of two parallel groups [29], one of them with low saturated fatty acids (SFA) (~7% total energy with red-, white- and non-meat SFA) versus a high SFA (~14% total energy, with the same food groups), was performed. Those diets high in SFA resulted in higher plasma total cholesterol, LDLc and non-HDLc than those low in SFA, independently of the animal- or plant-SFA origin. However, the authors did not compare the beginning of the three intervention periods, which were separated between 2 to 7 weeks. Moreover, the time that each participant was in the free diet during a washout period could influence the baseline status of each intervention period and also the observed differences at the end of the intervention periods. In this sense, in our study, both white- and red-meat periods had 8 weeks of duration with blood parameters at the beginning and end of each period. In addition, the inclusion of institutionalized adults lets us have a higher control on the overall diet of the participants. 

A recent meta-analysis concluded that the relationship between meat and the risk of stroke may differ by the type of consumed meat [30]. The results showed an association between total, red, processed and white meat consumption and total stroke incidence, with the odds ratios of 1.18 (1.09–1.28), 1.11 (1.03–1.20), 1.17 (1.08–1.25), and 0.87 (0.78–0.97), respectively. However, these results could be influenced by the fact that the meta-analysis was done by combining different food consumption units, such as grams/day or servings/day, as all the included results should be in the same units. Additional aspects, such as the type of meat consumed, i.e., lean or fatty pieces, or cooking methods, should be assessed in future meta-analyses in order to check their effect.

Most of the published studies did not differentiate between the types of meat consumed. In this sense, a recent review highlighted the different impacts on human health after the consumption of the same foods (meat, eggs or dairy) from animals raised differently [31]. No meta-analysis was performed due to the poor quality of the available information. Moreover, another additional source of variation is the husbandry system; in this sense, a recent systematic review with meta-analysis concluded that significant differences could be found when comparing organic and conventional meat in terms of fatty acid profiles [32]. Concentrations of SFA and MUFA were similar or slightly lower in organic compared with conventional meat, and larger differences were found in total PUFAs and *n*-2 PUFAs, which were 23% and 47% higher in organic meat. However, a higher heterogeneity was found, which could be explained by differences between animal species/meat types or the livestock conditions. In fact, grazing increases the incorporation of *n*-3 PUFAs into the muscle, even in ruminants that hydrogenate unsaturated fatty acids at rumen level due to the high content of fresh forage [33]. Cereals, on the other hand, show a higher composition of *n*-6 PUFA that will also be reflected in the meat composition, increasing the ratio of *n*-6/*n*-3 in the muscle [34].

### 4.3. Strengths and Limitations

The intervention design is an important strength as it provides a high level of scientific evidence. Firstly, the inclusion of a young adult population without diseases, which had a stable lifestyle pattern following the recommendations. In this sense, dietary records also indicated good compliance with the intervention recommendations, reinforcing the validity of the results. Moreover, the design allows for the comparison of the obtained results with themselves. The inclusion of a large set of blood parameters analyzed, including an exhaustive list of cardiovascular risk factors and fatty acid profiles is another strength.

On the other hand, the study also has several limitations. The current results are applicable to one specific type of meat, the Pirenaica breed, however, this kind of beef is similar to the most common husbandry conditions in the mountain production system in Spain. We couldn’t guarantee the independent effect and rule out the carry-over, the order or the interaction effect in all the variables, and some of the parameters could only be analyzed in the first period of the study.

## 5. Conclusions

In conclusion, the current study shows a similar body composition, cardiovascular risk factors and fatty acid profile response when consuming lean beef (Pirenaica breed) or chicken-based diets. These findings suggest that not all red meats are equal in terms of lipid effects and their cardiovascular risk associations. Extensive meat production systems, which guarantee animal welfare and good-production practices, could be incorporated into dietary counselling as a part of a healthy and sustainable diet, including an overall adequate distribution of protein intake from different sources (animal and plant-based). Specific dietary recommendations should be adapted to local consumers and products in order to enhance long-term compliance and also facilitate the guarantee of regional food systems that can maintain the local economy and achieve adequate social development in rural areas, which is essential for a sustainable food system.

## Figures and Tables

**Figure 1 nutrients-14-03724-f001:**
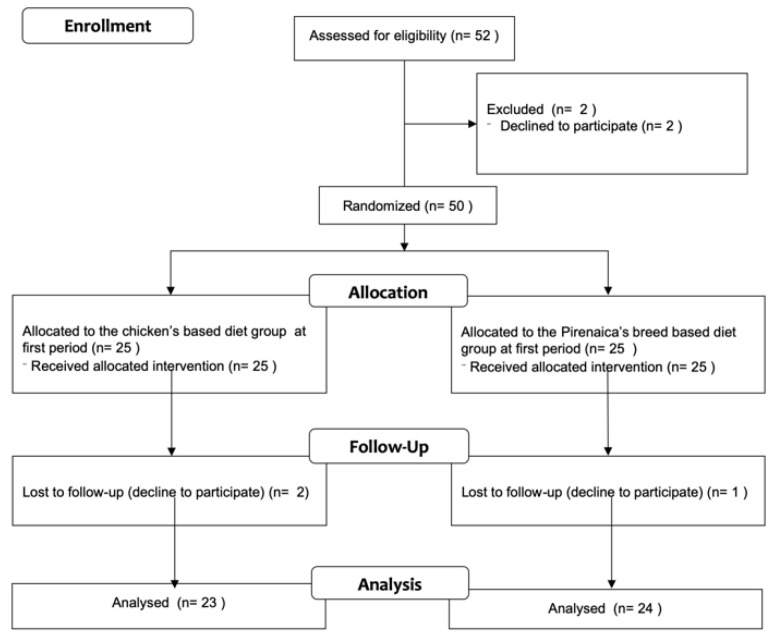
Flow diagram of the recruitment process.

**Figure 2 nutrients-14-03724-f002:**
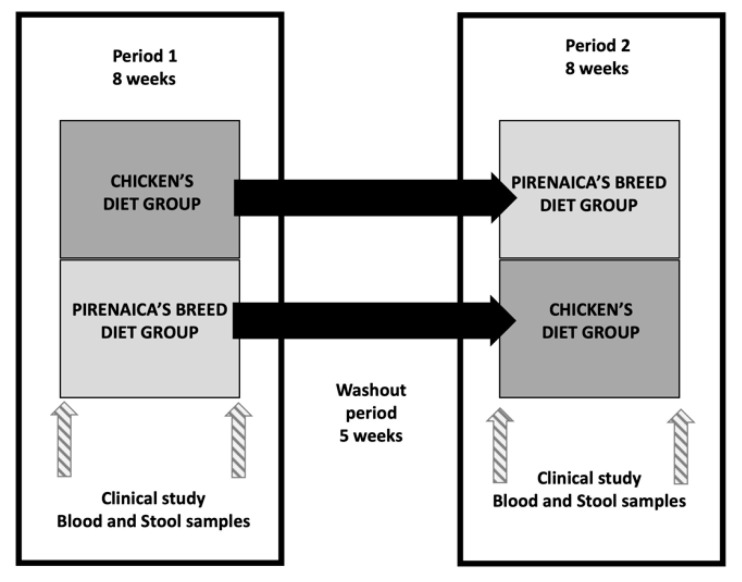
Study design.

**Table 1 nutrients-14-03724-t001:** Cooking methods characteristics for both beef and chicken.

	Grilled	Breaded and Deep Fried	Stewed
Olive oil	10 mL virgin olive oil	30 mL virgin olive oil	10 mL virgin olive oil
Cooking method	Simple grill	Fried	Stew
Internal temperature	75 °C	75 °C	75 °C
Cooking temperature	200 °C	200 °C	200 °C
Time of cooking	60 min	90 min	120 min
Additional foods Both recipes	Onions/Salt/black pepper/Garlic	Eggs/potatoes/bread crumb/black pepper/salt/herbs de Provence	Onions/water/tomatoes/white wine/mushrooms/carrots/salt/pepper/almonds/parsley/garlic/salt
Beef recipes	Meat broth/brandy/cooking cream/butter		Beef broth (beef, onions, carrots, leeks and water)
Chicken recipes	Tomatoes/red and green peppers/white wine/vinegar		Chicken broth (chicken, onions, carrots, leeks and water)
Beef cut Chicken cut	Loin Thigh and drumstick	Silverside Breast	Brisket Thigh and drumstick
Skin on chicken	Yes	No	Yes

**Table 2 nutrients-14-03724-t002:** Sociodemographic characteristics of included participants at baseline.

		Total Sample *n* = 47	Participants Who Started with the Pyrenees’ Beef Diet *n* = 24	Participants Who Started with the Chicken’s Diet *n* = 23	*p*
Gender *n* (%)	Female	23 (48.9)	7 (29.2)	16 (69.6)	0.006
	Male	24 (51.1)	17 (70.8)	7 (30.4)	
Age (years ± SD)		19.9 ± 1.75	20.5 ± 2.09	19.3 ± 1.02	0.022
Maternal education, *n* (%)	Low	10 (21.3)	4 (16.7)	6 (26.1)	0.691
	Medium	25 (53.2)	14 (58.3)	11 (47.8)	
	High	12 (25.5)	6 (25.0)	6 (26.1)	
Paternal education, *n* (%)	Low	13 (28.9)	6 (25.0)	7 (33.3)	0.814
	Medium	20 (44.4)	11 (45.8)	9 (42.9)	
	High	12 (26.7)	7 (29.2)	5 (23.8)	
BLOOD SAMPLES at baseline					
Hematology	Hemoglobin (106/μdL)		15.18 (1.12)	14.7 (0.94)	1.000
	Hematocrit (%)		44.61 (2.93)	43.9 (2.77)	0.062
Biochemistry	Glucose (mg/dL)		80.5 (5.87)	80.1 (4.10)	0.965
	GOT (U/L)		27.96 (10.58)	22.96 (5.10)	0.217
	GPT (U/L)		25.71 (10.98)	19.57 (4.99)	0.684
	GGT (U/L)		17.33 (8.64)	13.65 (2.85)	0.193
	Total cholesterol (mg/dL)		151.08 (22.85)	148.04 (41.57)	0.946
	Triglycerides (mg/dL)		72.79 (30.63)	62.30 (18.38)	0.286
	HDL cholesterol (mg/dL)		53.13 (13.02)	58.04 (16.44)	0.976
	LDL cholesterol (mg/dL)		83.33 (21.32)	77.66 (33.94)	0.726
Iron metabolism	Iron (μg/dL) (*n* = 46)		112.42 (46.33)	95.73 (35.87)	0.167
	Ferritin (ng/mL)		88.56 (58.03)	55.75 (30.53)	0.472
	Transferrin (mg/dL)		242.38 (25.37)	261.43 (33.69)	0.491
	Transferrin Saturation Index (%)		36.71 (14.37)	29.02 (11.47)	0.049
Other minerals	Magnesium (mmol/L)		0.87 (0.06)	0.89 (0.08)	0.074
	Calcium (mg/dL)		10.22 (0.29)	10.10 (0.45)	0.971
	Zinc (μg/dL)		88.93 (9.32)	88.17 (13.97)	0.327
Other variables	Apolipoprotein A1 (mg/dL)		156.96 (21.22)	166.70 (29.42)	0.562
	Apolipoprotein B (mg/dL)		78.5 (17.00)	73.43 (24.87)	1.000
	C Reactive Protein (mg/dL)		0.35 (0.56)	0.22 (0.07)	0.837

**Table 3 nutrients-14-03724-t003:** Mean values of body composition and blood parameters by intervention and control group: before, after, and mean differences over time between both groups (*n* = 47).

	Chicken-Based Diet Group *n* = 47				Pirenaica Breed-Based Diet Group *n* = 47				Mean Differences between Beginning and End between Both Groups *n* = 47			
	Before	After	F	*p* ^1^	Before	After	F	*p* ^1^	Mean Differences	95% IC	F	*p* ^2^
**Body Composition**												
Body Mass Index (kg/m^2^)	23.3 ± 3.45	23.35 ± 3.27	0.278	0.600	23.4 ± 3.40	23.3 ± 3.32	2.011	0.163	0.19	−0.16; 0.55	1.170	0.285
Fat mass index (kg/m^2^)	15.1 ± 6.46	14.9 ± 6.34	0.073	0.788	15.2 ± 6.51	14.8 ± 6.20	2.924	0.094	0.22	−0.19; 0.63	1.143	0.291
Waist Circumference (cm)	76.4 ± 8.47	81.5 ± 36.28	0.998	0.323	76.9 ± 8.98	76.4 ± 8.27	3.492	0.068	5.71	−4.58; 16.00	1.249	0.270
Blood Samples												
Biochemistry												
GOT (U/L)	25.3 ± 7.82	26.0 ± 7.23	0.333	0.567	27.2 ± 9.1	25.7 ± 6.6	0.979	0.328	2.23	−1.65; 6.12	1.341	0.253
Triglycerides (mg/dL)	75.1 ± 38.77	79.7 ± 32.08	1.833	0.182	70.3 ± 26.28	81.3 ± 37.01	7.915	0.007	−6.09	−17.39; 5.22	1.176	0.284
HDL cholesterol (mg/dL)	56.9 ± 14.66	56.5 ± 13.83	0.110	0.741	56.9 ± 15.19	57.1 ± 14.51	0.016	0.901	−0.57	−4.38; 3.24	0.090	0.765
Total cholesterol/HDL cholesterol	2.7 ± 0.70	2.8 ± 0.74	2.640	0.111	2.8 ± 0.79	2.8 ± 0.77	0.633	0.430	0.04	0.14; 0.21	0.177	0.676
Iron metabolism												
Iron (μg/dL) (*n* = 46)	101.1 ± 39.6	97.3 ± 34.02	0.363	0.550	112.6 ± 44.09	99.0 ± 33.78	3.363	0.073	−4.81	−18.16; 8.53	0.970	0.330
Transferrin Saturation Index (%)	30.6 ± 11.33	28.87 ± 9.93	0.945	0.336	35.5 ± 14.39	29.6 ± 11.2	6.826	0.012	4.15	−1.18; 9.47	2.460	0.124
Other minerals												
Zinc (μg/dL)	89.4 ± 12.58	91.3 ± 12.57	1.994	0.165	91.1 ± 12.71	96.1 ± 22.31	3.017	0.089	−3.17	−8.93; 2.59	1.229	0.274
Other variables												
Apolipoprotein A1 (mg/dL)	161.6 ± 25.86	157.5 ± 21.31	2.427	0.126	160.1 ± 23.86	158.8 ± 25.42	0.239	0.627	−2.68	−9.75; 4.38	0.585	0.448
Fatty Acids profile												
Palmitic Acid (%)	21.9 ± 2.80	22.3 ± 2.59	1.105	0.299	22.3 ± 2.34	22.2 ± 2.91	0.115	0.736	0.51	−0.72; 1.73	0.692	0.410
Stearic Acid (%)	7.2 ± 1.00	7.4 ± 0.82	1.774	0.190	7.1 ± 1.02	7.2 ± 0.85	0.008	0.927	0.17	−0.35; 0.69	0.437	0.512
Oleic Acid (%)	20.0 ± 4.15	19.8 ± 3.4	0.139	0.711	19.2 ± 3.04	20.3 ± 3.91	4.376	0.042	−0.82	−2.12; 0.49	1.603	0.213
AA_W6 (%)	7.3 ± 1.99	6.7 ± 1.83	11.198	0.002	7.3 ± 1.53	6.9 ± 1.49	4.142	0.048	−0.20	−0.76; 0.37	0.493	0.486
EPA (%)	0.0 ± 0.0	0.03 ± 0.19	1.704	0.199	0.0 ± 0.0	0.04 ± 0.19	1.703	0.199	0.03	−0.04; 0.09	0.789	0.379
DHA (%)	1.4 ± 0.83	1.7 ± 1.78	1.085	0.303	1.3 ± 0.96	1.3 ± 0.80	0.021	0.885	0.27	−0.39; 0.93	0.681	0.414
SFA (%)	30.9 ± 2.91	31.6 ± 2.69	2.813	0.101	31.1 ± 2.17	30.9 ± 2.70	0.060	0.807	0.76	−0.52; 2.03	1.434	0.238
UFA (%)	69.1 ± 2.91	68.4 ± 2.69	2.813	0.101	69.0 ± 2.17	69.1 ± 2.70	0.060	0.807	−0.76	−2.03; 0.52	1.434	0.238
MUFA (%)	21.0 ± 4.31	20.8 ± 3.55	0.212	0.647	20.0 ± 3.20	21.1 ± 3.85	4.131	0.048	−0.84	−2.13; 0.46	1.704	0.199
DUFA (%)	39.4 ± 5.13	39.1 ± 4.04	0.171	0.681	40.3 ± 4.43	39.8 ± 3.91	0.604	0.441	−0.01	−1.54; 1.51	0.000	0.985
MUFA/DUFA (%)	0.55 ± 0.17	0.54 ± 0.13	0.306	0.583	0.51 ± 0.13	0.54 ± 0.14	2.227	0.143	−0.03	−0.08; 0.02	1.113	0.298
PUFA (%)	48.0 ± 4.73	47.5 ± 3.68	0.743	0.394	48.9 ± 4.38	48.0 ± 4.00	1.676	0.202	−0.02	−0.06; 0.02	0.965	0.332
MUFA/PUFA (%)	0.45 ± 0.13	0.44 ± 0.11	0.118	0.733	0.42 ± 0.10	045 ± 0.12	2.906	0.095	−0.02	−0.06; 0.02	0.965	0.332
PUFA *n*6 (%)	45.7 ± 7.61	45.0 ± 7.12	1.533	0.223	47.6 ± 4.37	45.9 ± 7.23	2.641	0.111	0.83	−1.80; 3.46	0.409	0.526
PUFA *n*3 (%)	1.37 ± 0.83	1.7 ± 1.79	1.043	0.243	1.3 ± 0.96	1.3 ± 0.82	0.003	0.959	0.30	−0.36; 0.95	0.836	0.366
PUFA > 18C *n*6 (%)	7.3 ± 1.99	6.7 ± 1.83	11.198	0.002	7.3 ± 1.53	6.9 ± 1.49	4.142	0.048	−0.20	−0.76; 0.37	0.493	0.486
PUFA > 18C *n*3 (%)	1.4 ± 0.83	1.7 ± 1.79	1.403	0.243	1.3 ± 0.96	1.3 ± 0.82	0.003	0.959	0.30	−0.36; 0.95	0.836	0.366
SFA/MUFA (%)	1.5 ± 0.43	1.6 ± 0.32	0.224	0.638	1.6 ± 0.29	1.5 ± 0.33	2.957	0.092	0.07	−0.08; 0.21	0.982	0.328

Abbreviations: kg, kilograms; m^2^, squared meters; cm, centimeters; dL, deciliter; U, international units; L, liter; mg, milligrams; μg, micrograms; %, percentage; AA-W6, Arachidonic Acid; EPA, Eicosapentaenoic Acid: DHA, Docosahexaenoic Acid; SFA, saturated fatty acids; UFA, unsaturated fatty acids; MUFA, monounsaturated fatty acids; DUFA, diunsaturated fatty acids; PUFA, polyunsaturated fatty acids. Lipid number: Palmitic Acid, C16:0; Stearic Acid, C18:0); Oleic Acid, C18:1n9; AA_W6, C20:4n6; EPA, C20:5n3; DHA, C22:6n3; F, F statistic; IC, confidence interval. ^1^ Multivariate contrast between the beginning and the end of each corresponding food product (chicken or beef). ^2^ Multivariate contrast between the mean differences (final—beginning) of both food products (chicken or beef) periods.

**Table 4 nutrients-14-03724-t004:** Mean values of body composition, blood parameters and mean differences over time between both groups in the first period of the study.

	Chicken-Based Diet Group *n* = 23				Pirenaica Breed-Based Diet Group *n* = 24				Mean Differences between Beginning and End between Control and Intervention in the First Period *n* = 47			
	Before	After	F	*p* ^1^	Before	After	F	*p* ^1^	Mean Differences	95% IC	F	*p* ^2^
**Blood Pressure**												
Systolic Pressure (mmHg)	109.99 ± 10.15	113.23 ± 9.82	5.94	0.023	116.04 ± 15.28	117 ± 13.95	0.728	0.402	1.927	−2.20; 6.05	0.885	0.352
Diastolic Pressure (mmHg)	66.81 ± 6.92	67.03 ± 6.80	0.032	0.860	67.19 ± 5.33	68.43 ± 6.18	0.866	0.362	−1.019	−4.66; 2.62	0.318	0.576
Blood Samples												
Hematology												
Hemoglobin (106/μdL)	109.99 ± 10.15	113.23 ± 9.82	1.694	0.002	116.04 ± 15.28	117 ± 13.95	3.473	0.075	−0.169	−0.47; 0.13	1.319	0.257
Hematocrit (%)	66.81 ± 6.92	67.03 ± 6.80	1.611	0.218	67.19 ± 5.33	68.43 ± 6.18	2.435	0.132	−1.025	−2.06; 0.01	4.000	0.052
Biochemistry												
Glucose (mg/dL)	80.13 ± 4.10	77.13 ± 5.83	8.496	0.008	80.46 ± 5.87	77.50 ± 6.90	6.573	0.017	−0.042	−3.17; 3.08	0.001	0.979
GPT (U/L)	19.57 ± 5.00	23.09 ± 10.3	2.673	0.116	25.11 ± 10.98	28.96 ± 11.91	1.741	0.200	0.272	−6.34; 6.88	0.007	0.934
GGT (U/L)	19.57 ± 5.00	23.09 ± 10.3	30.646	<0.001	25.11 ± 10.98	28.96 ± 11.91	20.748	<0.001	−1.697	−3.66; 0.26	3.037	0.088
Total cholesterol (mg/dL)	148.04 ± 41.57	149.22 ± 36.28	0.182	0.674	151.08 ± 22.85	152.71 ± 27.07	0.290	0.596	−0.451	−8.70; 7.80	0.012	0.913
LDL cholesterol (mg/dL)	77.65 ± 33.94	74.48 ± 31.34	2.073	0.164	83.33 ± 21.32	80.00 ± 21.35	2.249	0.147	0.159	−6.15; 6.47	0.003	0.960
LDL cholesterol/HDL cholesterol	1.40 ± 0.71	1.31 ± 0.61	1.842	0.188	1.70 ± 0.68	1.56 ± 0.57	4.903	0.037	0.05	−0.13; 0.24	0.367	0.548
Iron metabolism												
Ferritin (ng/mL)	55.75 ± 30.53	44.11 ± 45.70	1.296	0.267	88.56 ± 58.03	66.92 ± 47.01	24.707	<0.001	10.000	−12.05; 32.07	0.835	0.366
Transferrin (mg/dL)	261.43 ± 33.69	276.33 ± 30.01	12.811	0.002	242.38 ± 25.38	266.42 ± 33.26	31.398	<0.001	−9.146	−21.20; 2.91	2.336	0.133
Other minerals												
Magnesium (mmol/L)	0.89 ± 0.08	0.83 ± 0.05	9.625	0.005	0.87 ± 0.06	0.84 ± 0.05	3.119	0.091	−0.032	−0.08; 0.02	1.599	0.213
Calcium (mg/dL)	10.10 ± 0.45	9.52 ± 0.40	30.985	<0.001	10.22 ± 0.29	9.46 ± 0.26	123.626	<0.001	0.167	−0.08; 0.42	1.810	0.185
Other variables												
Apolipoprotein B (mg/dL)	73.43 ± 24.87	69.61 ± 19.91	4.738	0.041	78.50 ± 17.00	76.21 ± 16.06	2.510	0.127	−1.534	−6.10; 3.03	0.458	0.502
C Reactive Protein (mg/dL)	0.22 ± 0.08	0.51 ± 0.09	134.096	<0.001	0.34 ± 0.56	0.52 ± 0.14	2.244	0.148	0.117	−0.13; 0.36	0.929	0.340
Fatty Acids												
Linoleic Acid (%)	38.01 ± 3.34	39.36 ± 4.72	2.005	0.171	39.74 ± 4.72	39.78 ± 3.87	0.002	0.961	1.308	1.20; 3.81	1.11	0.298

Abbreviations: mmHg, millimeter of mercury; μdL, micro deciliter; %, percentage; mg, milligrams; dL, deciliter; U, international units; L, liter; ng, nanograms; mmol, millimole; F, F statistic; IC, confidence interval. Lipid number: Linoleic Acid, C18:2 n6. ^1^ Multivariate contrast between the beginning and the end of each corresponding food product (chicken or beef). ^2^ Multivariate contrast between the mean differences (final—beginning) of both food products (chicken or beef) periods.

**Table 5 nutrients-14-03724-t005:** Mean values of iron metabolism between both chicken and Pyrenees’ beef diet’s groups by gender, in the overall sample, and in the first period of the study.

Blood Samples Iron Metabolism		Chicken-Based Diet Group				Pyrenees Beef-Based Diet Group				Mean Differences between Beginning and End between Both Groups			
		Before	After	F	*p* ^1^	Before	After	F	*p* ^1^	Mean Diff	95% IC	F	*p* ^2^
**Overall sample (*n* = 47)**													
Iron (μg/dL) (*n* = 46)	Male	98.8 ± 35.30	102.5 ± 37.41	0.279	0.602	116.91 ± 43.55	104.96 ± 30.97	1.065	0.313	12.21	−18.78; 43.20	0.671	0.422
	Female	103.50 ± 44.37	91.78 ± 29.92	1.175	0.290	108.96 ± 30.97	94.13 ± 36.50	2.741	0.112	0.000	−24; 44; 24.44	0.000	1.000
Transferrin Saturation Index (%)	Male	30.53 ± 10.63	30.91 ± 10.25	0.035	0.852	39.47 ± 15.10	32.50 ± 10.63	3.239	0.085	7.86	1.99; 17.72	2.734	0.112
	Female	30.76 ± 12.25	26.73 ± 9.32	1.799	0.194	31.36 ± 12.62	26.50 ± 11.15	4.211	0.052	1.06	−5.79; 7.91	0.104	0.750
1st period of the study (Control: n = 23. Intervention. n = 24)													
Ferritin (ng/mL)	Male	65.13 ± 28.34	76.00 ± 72.69	0.125	0.736	111.42 ± 51.75	86.00 ± 41.86	20.218	<0.001	36.29	−7.44; 80.02	2.962	0.099
	Female	51.65 ± 31.41	30.16 ± 16.35	13.828	0.002	33.05 ± 26.31	20.57 ± 16.11	6.921	0.039	−9.00	−28.46; 10.46	0.925	0.347
Transferrin (mg/dL)	Male	257.14 ± 31.27	279.55 ± 34.72	8.216	0.029	236.12 ± 23.22	252.32 ± 25.26	21.192	<0.001	1.05	−12.22; 14.32	0.027	0.871
	Female	263.31 ± 35.52	274.91 ± 28.84	5.746	0.030	257.57 ± 25.51	300.66 ± 24.63	24.308	<0.001	6.74	−5.16; 18.63	1.387	0.252

Abbreviations: Mean dif, mean difference; μg, micrograms; dL, deciliter; %, percentage; ng, nanograms; mL, milliliter; F, F statistic; IC, confidence interval. ^1^ Multivariate contrast between the beginning and the end of each corresponding food product (chicken or beef). ^2^ Multivariate contrast between the mean differences (final—beginning) of both food products (chicken or beef) periods.

## Data Availability

Available information about the study will be included at http://dietapyr2.com, accessed on 6 September 2022.

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
