# Peer review of "Effect of the Intake of Lean Red-Meat from Beef-(Pirenaica Breed) versus Lean White-Meat on Body Composition, Fatty Acids Profile and Cardiovascular Risk Indicators: A Randomized Cross-Over Study in Healthy Young Adults"

_nutrients, 2022, doi:10.3390/nu14183724_

Round 1

Reviewer 1 Report

In this manuscript the authors aimed to assess the effect of the consumption of beef- (from Pirenaica breed) or chicken- based diets on body composition, fatty acids profile  and CV risk indicators in healthy adults. The study is well designed however some points should be addressed:

1)    Why it was decided a period of 8-week follow up? What was the reason behind not considering longer follow-up period? This should be justified in paper since it may influence the observed effect of each diet.

2)    Differences of beef or chicken –based diets on CV risk gain a value mainly in patients with cardiovascular risk factors. Therefore, why did the authors had excluded this category of subjects with CV risk factors.

3)    The beef was from Pirenaica breed, hence which types of beef may lead to  similar observed results.

4)     no differences were found in body composition, fatty acid profile or CV risk indicators from baseline in either diet group, here raises important question which is weather the study population belong to same race (Caucasian, white, etc…). This point is essential due to the well know differences in the protein/lipid metabolism between races.

5)    What was the level of pro-and antiinflammtory cytochines (es.TNF, IL6, IL10). This is essential to determine understand the CV impact of each diet

Author Response

Answer to the reviewer’s comments 

Nutrients

No. nutrients-1869283

Dear Editor,

We would like to submit a revised version of our manuscript “Effect of the Intake of Lean Red-meat from Beef- (Pirenaica Breed) versus Lean White-meat on Body Composition, Fatty Acids Profile and Cardiovascular Risk Indicators: a Randomised Cross-over Study in Healthy Young Adults”. We would like to thank the possibility to review the manuscript and you can find the revised manuscript which included the reviewer’ suggestions. Changes in the original manuscript are highlighted in yellow.

#Reviewer 1. Comments and Suggestions for Authors

In this manuscript the authors aimed to assess the effect of the consumption of beef- (from Pirenaica breed) or chicken- based diets on body composition, fatty acids profile  and CV risk indicators in healthy adults. The study is well designed however some points should be addressed:

  • Why it was decided a period of 8-week follow up? What was the reason behind not considering longer follow-up period? This should be justified in paper since it may influence the observed effect of each diet.

Answer

Thanks for the comment. The design of the working 8-week period was based in several reasons. Firstly, in previous researches developed with the same design (reference number 24 of the manuscript). Also, the population group (healthy young adults) imply the necessity to working in accommodation halls in order to facilitate the food consumed products in standardized way (food frequency, cooking methods, etc.). The University year in Spain was based in 3 trimesters, and we need to adapt the study design with the university periods in order to perform the clinical study and also to guarantee the consumption of the products. In this sense, the accommodation halls give us the opportunity to control the consumption of the products.

An additional paragraph has been added in order to facilitate the explanation for the readers in lines 115 to 123. “The study design and duration of the study was based on previous research [26]. Also, the population group (young healthy adults) implies the need to work in university accommodation halls to facilitate the consumption of food in a standardized way (frequency of food, cooking methods, etc.). Moreover, the University course in Spain is based on 3 trimesters, and we need to adapt the study design with university periods to perform the clinical study in the periods of attendance of participants, and also to guarantee the consumption of products. Additionally, residences give us the opportunity to control the consumption of products (chicken or Pirenaica’s breed at lunch time) by the accommodation hall's personal.”

2)    Differences of beef or chicken –based diets on CV risk gain a value mainly in patients with cardiovascular risk factors. Therefore, why did the authors had excluded this category of subjects with CV risk factors.

 Answer

Thanks for the suggestion. The hypothesis was working firstly with healthy subjects in order to test the effect on this population group, in order not to have any potential confounding factor. Afterwards, the next step will be to test the effect on people with several CV risk factors.

3)    The beef was from Pirenaica breed, hence which types of beef may lead to  similar observed results.

Answer

Thanks for the comment. An indication has been added towards some characteristics of this breed (lines 73 to 74) since it is a beef orientated breed with a fast growth and a low intramuscular fat content. These aspects, together with the production system grazing in a mountainous area, define the type of animal.

4)    No differences were found in body composition, fatty acid profile or CV risk indicators from baseline in either diet group, here raises important question which is weather the study population belong to same race (Caucasian, white, etc…). This point is essential due to the well know differences in the protein/lipid metabolism between races.

Answer

Thanks for the suggestion. We have included the race in the participant description section.

5)   What was the level of pro-and antiinflammtory cytochines (es.TNF, IL6, IL10). This is essential to determine understand the CV impact of each diet

Answer

Thank you for the comment. As you suggested we included the baseline parameters of all the blood simple in Table 2 in order to see that no differences were observed at the beginning of the project.

Reviewer 2 Report

Summary of Manuscript: The aim of this study was to determine whether the consumption of beef- (Pirenaica breed) or chicken- based diets impact body composition, fatty acid profiles, and cardiovascular risk factors. This was a randomized, cross-over, controlled trial in which the participants consumed each diet for 8 weeks each, with a 5-week washout period between diets. The results demonstrated no significant differences between diets regarding these cardiovascular risk factors.

Broad comments: I carefully reviewed this manuscript. The authors provided an interesting study; however, my primary concern is the study design. In addition, please check spelling and grammar throughout the manuscript. I provided specific comments below.

Abstract

Point 1: Line 29: Should for a 8-week periods” be changed to “for 8-week periods”?

Point 2: Line 32: Should “control groups comparisons” be changed to “control group comparisons”?

Point 3: Line 33: It’s not recommended to begin sentences with numbers. Therefore, “47” should be spelled out.

Introduction

Point 4: Line 55: Should “meta-analysis” be changed to “meta-analyses”?

Point 5: Line 68: Should “taking” be changed to “taken”?

Point 6: Line 76: Should “choice the type of veal,” be changed to “choice for the type of veal” or similar change?

Point 7: Lines 77-80: Please revise this sentence. It is difficult to follow.

Materials and Methods

Point 8: Line 95: Figure 1 legend: Please revise the following: “This is a figure. Schemes follow the same formatting.”

Point 9: Line 102: Should “stool sample were” be changed to “stool samples were”?

Point 10: Line 105: Please include a Figure 2 legend.

Point 11: Lines 111-113: Please revise this sentence. It is difficult to understand.

Point 12: Line 119: Should “up 20%” be changed to “up to 20%”?

Point 13: Line 123: Should “was the responsible to enrolled participants” be changed to “was responsible to enroll participants”?

Point 14: Line 124: Should “18,1 to 27,5” be changed to “18.1 to 27.5”?

Point 15: Line 127: Should “before start the baseline” be changed to “before starting the baseline” or similar change?

Point 16: Line 145: Is “hells” spelled correctly?

Point 17: Line 162: Why was a food frequency questionnaire selected rather than 24-hour recalls? Was a dietary analysis program used to analyze nutrients consumed (macronutrients, vitamins, minerals, etc.)?

Point 18: Lines 169-170: Is it possible to include a table of the nutritional compositions for both diets?

Point 19: Line 176: Should “designed” be changed to “designated”?

Point 20: Line 182: Please change “Sample” to “Samples”

Point 21: Line 189: Please add more details for VITROS500 and SYSMEX.

Point 22: Lines 197-198: Are “Eicosapentadienoic Acid” and “Docosahexadienoic Acid” spelled correctly?

Point 23: Line 226: Should “social” be capitalized?

Results

Point 24: Line 230-231: What does it mean that they “did not follow the diet exactly as offered in the accommodation halls, but made acceptable modification from diet)”? Did some participants not follow the diets?

Point 25: Table 2: Please explain low, medium, and high education.

Point 26: Lines 242-243: Please clarify the following sentence: “no statistical differences were observed in none of the body composition or blood sample parameters.”

Point 27: Line 245: Is “complain” the correct word?

Point 28: Line 257: Is “observer” the correct word?

Point 29: Line 280: Should “sample” be plural?

Discussion

Point 30: Line 340: Should “compared” be changed to “compare”?

Point 31: Lines 345-346: It is stated that “In addition, the inclusion of institutionalized adults, let us have a higher control on the overall diet of the participants.” What control did you have on the overall diets of the participants? The participants were asked to consume a “healthy” diet or to consume their “usual” diet. Were the participants instructed to consume specific food items?

Point 32: Line 358: Please change “egss” to “eggs”

Point 33: Line 369: “[32.” is missing a bracket.

Point 34: Line 371: Is “reducing the ratio n-6/n-3 in the muscle” correct?

Point 35: Line 373: Should “provide” be changed to “provides”?

Conclusions

Point 36: Line 391: Please change “good-productions practices” to “good-production practices”

Point 37: Line 393: Is “vegetal” spelled correctly?

Point 38: Line 396: Please revise the following: “systems which let to develop socio-cultural adaptations.” It is difficult to understand.

Point 39: Are there significant nutrient or kilocalorie differences between the intervention (beef) and control (chicken) groups? For example, if there are differences in macro- and micronutrient intakes, how can conclusions be made between these two diets? There should be a table comparing the consumption of kilocalories and nutrients between both diets.

Point 40: It should also be noted in the manuscript that Body Mass Index does not measure body composition. 

Author Response

Answer to the reviewer’s comments 

Nutrients

No. nutrients-1869283

Dear Editor,

We would like to submit a revised version of our manuscript “Effect of the Intake of Lean Red-meat from Beef- (Pirenaica Breed) versus Lean White-meat on Body Composition, Fatty Acids Profile and Cardiovascular Risk Indicators: a Randomised Cross-over Study in Healthy Young Adults”. We would like to thank the possibility to review the manuscript and you can find the revised manuscript which included the reviewer’ suggestions. Changes in the original manuscript are highlighted in yellow.

#Reviewer 2. Comments and Suggestions for Authors

 Summary of Manuscript: The aim of this study was to determine whether the consumption of beef- (Pirenaica breed) or chicken- based diets impact body composition, fatty acid profiles, and cardiovascular risk factors. This was a randomized, cross-over, controlled trial in which the participants consumed each diet for 8 weeks each, with a 5-week washout period between diets. The results demonstrated no significant differences between diets regarding these cardiovascular risk factors.

Broad comments: I carefully reviewed this manuscript. The authors provided an interesting study; however, my primary concern is the study design. In addition, please check spelling and grammar throughout the manuscript. I provided specific comments below.

Thank you for the overall comments.

Abstract

Point 1: Line 29: Should “for a 8-week periods” be changed to “for 8-week periods”?

 Answer

Thanks for the suggestion. Changed accordingly.

Point 2: Line 32: Should “control groups comparisons” be changed to “control group comparisons”?

Answer

Thanks for the suggestion. Changed accordingly.

Point 3: Line 33: It’s not recommended to begin sentences with numbers. Therefore, “47” should be spelled out.

 Answer

Thanks for the suggestion. We have change the beginning of the sentence accordingly.

Introduction

Point 4: Line 55: Should “meta-analysis” be changed to “meta-analyses”?

Answer

Thanks for the suggestion. Changed accordingly.

Point 5: Line 68: Should “taking” be changed to “taken”?

Answer

Thanks for the suggestion. Changed accordingly.

Point 6: Line 76: Should “choice the type of veal,” be changed to “choice for the type of veal” or similar change?

Answer

Thanks for the suggestion. Changed accordingly.

Point 7: Lines 77-80: Please revise this sentence. It is difficult to follow.

Answer

Thanks for the comment. The sentence has been revised accordingly.

Materials and Methods

Point 8: Line 95: Figure 1 legend: Please revise the following: “This is a figure. Schemes follow the same formatting.”

Answer

Thanks for the suggestion. The legend has been included

Point 9: Line 102: Should “stool sample were” be changed to “stool samples were”?

Answer

Thanks for the suggestion. Changed accordingly.

Point 10: Line 105: Please include a Figure 2 legend.

Answer

Thanks for the suggestion. The legend has been included

Point 11: Lines 111-113: Please revise this sentence. It is difficult to understand.

Answer

Thanks for the comment. The sentence has been revised accordingly.

Point 12: Line 119: Should “up 20%” be changed to “up to 20%”?

Answer

Thanks for the suggestion. Changed accordingly.

Point 13: Line 123: Should “was the responsible to enrolled participants” be changed to “was responsible to enroll participants”?

 Answer

Thanks for the suggestion. Changed accordingly.

Point 14: Line 124: Should “18,1 to 27,5” be changed to “18.1 to 27.5”?

Answer

Thanks for the suggestion. Changed accordingly.

Point 15: Line 127: Should “before start the baseline” be changed to “before starting the baseline” or similar change?

Answer

Thanks for the suggestion. The verb has been changed.

Point 16: Line 145: Is “hells” spelled correctly?

Answer

Thanks for the comment. The work has been deleted.

Point 17: Line 162: Why was a food frequency questionnaire selected rather than 24-hour recalls? Was a dietary analysis program used to analyze nutrients consumed (macronutrients, vitamins, minerals, etc.)?

Answer

Thank you very much for the suggestion. The reason behind the selection of the food frequency questionnaire was to analyze the overall effect of the diet the previous months  before starting the baseline of the project, and also having a tool to check the evolution across both periods of the project. With the 24-h dietary recall we only have the information about the previous day and we considered that was not the most appropriate tool in our case.

Regarding the nutrients consumed, we did not performed any kind of analysis, therefore no dietary analysis program was used.

Point 18: Lines 169-170: Is it possible to include a table of the nutritional compositions for both diets?

Answer

Thanks for the comment. The participants were instructed to consume their usual diet I order to maintain the personal habits and only modify it with the nutritional intervention product. They only need to include 3 times per week the intervention (Pirenaica’s breed) or the control (chicken) product. In this sense, no specific diet were applied to each group. We have included a sentence in the intervention section between lines 181-182.

Point 19: Line 176: Should “designed” be changed to “designated”?

 Answer

Thanks for the suggestion. The verb has been changed.

Point 20: Line 182: Please change “Sample” to “Samples”

 Answer

Thanks for the suggestion. Changed accordingly.

Point 21: Line 189: Please add more details for VITROS500 and SYSMEX.

Answer

Thanks for the suggestion. Due to the huge amount of techniques used with the blood samples and the space limitation we have maintain the current version of this section.

Point 22: Lines 197-198: Are “Eicosapentadienoic Acid” and “Docosahexadienoic Acid” spelled correctly?

Answer

Thank you for the comment. Both acids has been corrected through the manuscript.

Point 23: Line 226: Should “social” be capitalized?

 Answer

Thanks for the suggestion. Changed accordingly.

Results

Point 24: Line 230-231: What does it mean that they “did not follow the diet exactly as offered in the accommodation halls, but made acceptable modification from diet)”? Did some participants not follow the diets?

 Answer

Thanks for the comment. The participants were instructed to follow their usual diet in the accommodation hall, including 3 times per week the chicken or the Pirenaica’s breed at lunch time. The personnel of the of accommodation hall controlled at lunch time that the participants consumed their corresponding product. Also, the evaluation of their diet over time were analyzed using a diet quality index, which showed no differences between the beginning and the end of each period, and also between control and intervention periods (data not published yet). In this sense we have included a sentence in the study design section (lines 121 to 123) and in the intervention section (lines 181-182)

Point 25: Table 2: Please explain low, medium, and high education.

 Answer

Thank you very much for the comment. A new section with the information about the these variables has been included in lines 144-148.

Point 26: Lines 242-243: Please clarify the following sentence: “no statistical differences were observed in none of the body composition or blood sample parameters.”

 Answer

Thanks for the comment. The sentence has been clarified.

Point 27: Line 245: Is “complain” the correct word?

 Answer

Thanks for the suggestion. The verb has been changed.

Point 28: Line 257: Is “observer” the correct word?

 Answer

Thanks for the suggestion. Changed accordingly.

Point 29: Line 280: Should “sample” be plural?

Answer

Thanks for the suggestion. Changed accordingly.

Discussion

Point 30: Line 340: Should “compared” be changed to “compare”? 

Answer

Thanks for the suggestion. The verb has been changed.

Point 31: Lines 345-346: It is stated that “In addition, the inclusion of institutionalized adults, let us have a higher control on the overall diet of the participants.” What control did you have on the overall diets of the participants? The participants were asked to consume a “healthy” diet or to consume their “usual” diet. Were the participants instructed to consume specific food items?

Answer

Thanks for the comment. An additional paragraph has been included in the study design section in lines 121-123. Also, a clarification about the nutritional intervention has been included in lines 181-182.

Point 32: Line 358: Please change “egss” to “eggs”

 Answer

Thanks for the suggestion. Changed accordingly.

Point 33: Line 369: “[32.” is missing a bracket.

 Answer

Thanks for the suggestion. Changed accordingly.

Point 34: Line 371: Is “reducing the ratio n-6/n-3 in the muscle” correct?

 Answer

Thank you very much for the comment. We have reviewed the reference and changed the sentence accordingly.

Point 35: Line 373: Should “provide” be changed to “provides”?

Answer

Thanks for the suggestion. The verb has been changed.

Conclusions

Point 36: Line 391: Please change “good-productions practices” to “good-production practices”

 Answer

Thanks for the suggestion. Changed accordingly.

Point 37: Line 393: Is “vegetal” spelled correctly?

 Answer

Thanks for the suggestion. Changed accordingly.

Point 38: Line 396: Please revise the following: “systems which let to develop socio-cultural adaptations.” It is difficult to understand.

 Answer

Thank you very much for the suggestion. The sentence has been clarified in lines 402 to 405: “Specific dietary recommendations should be adapted to local consumers and products, in order to enhance long-term compliance, and also facilitate the guarantee of regional food systems that can maintain the local economy and achieve adequate social development in rural areas, essential for a sustainable food system.”

Point 39: Are there significant nutrient or kilocalorie differences between the intervention (beef) and control (chicken) groups? For example, if there are differences in macro- and micronutrient intakes, how can conclusions be made between these two diets? There should be a table comparing the consumption of kilocalories and nutrients between both diets.

 Answer

Thank you for the question. As we commented before, no nutritional analysis were made in terms of macro – or micro- nutrient intake. We checked the differences in terms of frequency of food and beverages and we analyses the differences using an index (diet quality index). We consider a good approach to evaluate the overall quality of the diet due to the participants were instructed to consume the usual diet, and no control in terms of macro- and micro- nutrient intake. In this sense the results showed that no differences in terms of food and beverage consumption was observed between participants at both periods (data not published yet), therefore the participants maintain their usual habits across the  duration of the project.

Point 40: It should also be noted in the manuscript that Body Mass Index does not measure body composition. 

Answer

Thank you for the comment. Traditionally the body mass index has been used as the easiest body composition indicator, however we agree with you that is not the best option. In order to group the indicators, we decided to name them as body composition indicators in order to facilitate the information for the readers.

Round 2

Reviewer 1 Report

The authors addressed my comments. The paper in its present for is improved.